# CLMIU: Commonsense Learning in Multimodal Image Understanding

## Abstract

The problem of automatically describing the content of an image through accurate and meaningful captions has been attracting considerable attention among computer vision researchers. Recently, Transformers have been applied to image captioning to encode cross-modal information, in conjunction with Convolutional Neural Networks, which provide image region descriptions in terms of embeddings and object labels as input. However, the generated captions sometimes fail to capture the intentions, relationships, and abstract concepts that rely on general or commonsense knowledge. In this work we propose a novel network design, combining the strengths of Transformer models with graph-based models conveying external (common sense) knowledge. Our proposed architecture is a pure vision transformer-based image captioning model, with sequences of image patches used directly as input, without extracting any regional features. In particular, unlike the prior work, our architecture incorporates a knowledge-augmented encoder with a Transformer backbone to inject the external knowledge extracted from a knowledge graph. Furthermore, the bidirectional training on a vision-language corpus of image-text pairs, using modality specific self-supervised learning objectives, achieves promising results compared to the state-of-the-art. Our method has been trained from scratch on a small dataset, achieving a 3.8%, 2.7%, 3.2% and 6.3% improvement in BLEU@4, Meteor, Rouge and Cider scores respectively. We also reported competitive results on the NoCaps dataset, showing that the model generalizes to unseen object categories.

## 1 Introduction

Image captioning (IC) is an important research area of Computer Vision (CV) which addresses the problem of automatically describing the content of an image. The generated description includes the global scene, the objects contained in the image, their relationships as well as their attributes and the activities they are involved in. Training multimodal models on manually annotated paired image and text corpora aims to learn cross-modal representations that capture rich image and language semantics. Factual and commonsense knowledge are essential to how humans understand the world around them and learn about it. Factual knowledge refers to the specific details or elements of a subject (e.g. *"London is the capital of the United Kingdom"*). Commonsense knowledge includes information about events and their effects, about physical objects and how they are perceived, and about their properties and their relations to one another McCarthy et al. (1960). A large amount of this knowledge is common to all humans, hence the term "common" in "common-sense".

Commonsense knowledge is hard to compute/learn by machine learning models. Therefore, incorporating commonsense information is at present a key problem facing machine learning research Klein & Nabi (2019); Zhou et al. (2019); Zhang et al. (2019); Wang et al. (2020); Liu et al. (2020). Even the state-of-the-art (SOTA) models in image captioning ignore this type of knowledge Li et al. (2019a;b); Lu et al. (2019); Tan & Bansal (2019); Chen et al. (2020); Desai & Johnson (2020); Li et al. (2020b); Hu et al. (2020); Zhang et al. (2021b). Though some captions might hint at learning elaborated abstract concepts, it is not obvious that even training on, e.g., 1.8B image/text pairs Wang et al. (2021) will result in models capable of acquiring commonsense knowledge. We argue that using exclusively pre-trained language models and the concepts learned by them cannot provide sufficient information for image captioning. Incorporating external commonsense knowledge into the image captioning methods relies primarily on the intuition that human beings produce

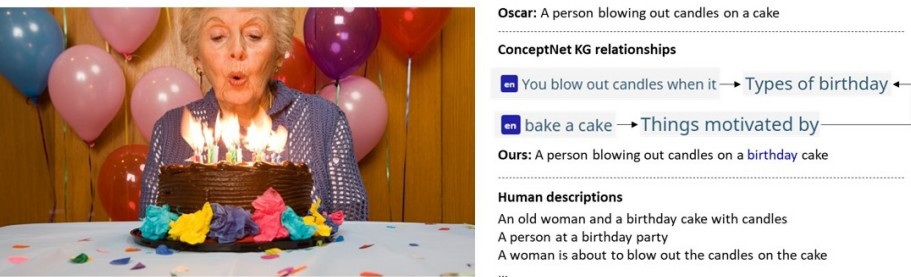

Figure 1: This is an example image showing a case when identifying semantic concepts, not explicitly represented in the scene, would help to provide a better description. Commonsense knowledge is used to relate the elements, namely people, cake and candle, to the concept of a birthday. The comparison between the baseline caption and the one generated by our method shows the benefit of incorporating external knowledge. The relationship information is extracted from the ConceptNet Speer et al. knowledge graph.

accurate and semantically rich descriptions of scenes by exploiting this source of information. In Figure 1, it can be seen that the woman is getting ready to blow out the candles on the cake. We can infer that the occasion might be a birthday celebration just by looking at the cake, the candles and the overall composition of the scene. This is the type of knowledge that is very relevant for the task of captioning. It is typically encoded in a Knowledge Graph (KG) in the form of nodes representing objects and their relationships. As can be seen in the figure, the incorporation of external knowledge from a knowledge graph improves the caption by suggesting the semantic concept of 'birthday'. Furthermore, current vision and language models often leverage a frozen object detector, like Faster R-CNN, pre-trained on labelled object detection datasets. Such approaches are limited by the granularity of these datasets, and so, are less scalable.

In this paper, we introduce CLMIU, short for **C**ommonsense **L**earning in **M**ultimodal **I**mage **U**nderstanding. CLMIU's objective is to incorporate factual and commonsense knowledge in a multimodal network by training on a small dataset with access to external information. Aiming to learn accurate object-detector free image descriptions beyond explicit image content, we train CLMIU to **a)** recover masked language and transformed vision tokens from its contextualized vector representations, and to **b)**, identify the correct alignment of the textual concepts to their corresponding image regions in the image-text pairs. We show both quantitatively and qualitatively that CLMIU has a strong understanding of factual and commonsense elements needed to accurately describe natural images. When using considerably less image-text data for training, CLMIU outperforms strong baselines like Oscar Li et al. (2020b), VinVL Zhang et al. (2021b) and Vilt Kim et al. (2021). Specifically, our method achieves a 3.8%, 2.7%, 3.2% and 6.3% improvement in BLEU@4 Papineni et al. (2002), METEOR Denkowski & Lavie (2014), Rouge Lin (2004) and CIDEr Vedantam et al. (2015) scores respectively on the MS COCO Captions dataset Chen et al. (2015). This improvement in caption generation emerges during pre-training with an external knowledge source. The results obtained on the NoCaps Agrawal et al. (2019) dataset show that the model generalizes to unseen object categories. An analysis of CLMIU's attention patterns (supplementary material Appendix Figure 4) shows that image regions attend to text tokens that are conceptually related.

Finally, ablation studies of CLMIU show that 1) external knowledge injection works better when added to the last layer(s) rather than from the start of the multimodal encoder, 2) existing methods improve by re-training with the external knowledge encoder, 3) using Group Mask Model Learning (GMML) Atito et al. (2021b) for the vision encoding leads to better image representations and 4) CLMIU's performance steadily improves by training longer. The combination of these results suggests that incorporating factual and commonsense knowledge into image captioning models is a promising path forward for future research. In summary, our main contributions are:

1. CLMIU, a performant end-to-end vision and language model, that learns multimodal image representations from images and their captions incorporating external knowledge. To the best of our knowledge, CLMIU is the first to propose the use of an external knowl-

edge graph to learn knowledge-informed multimodal image and text representations using transformers.

2. A detector-free multimodal model able to produce multi-concept image representations by leveraging the strengths of GMML.

3. A set of experiments/ablation studies demonstrating the strong performance of CLMIU in image captioning. It improves the performance of VLP models by producing semantically better captions.

## 2 RELATED WORK

**Vision-Language Pre-training.** Inspired by the success of large scale pre-trained language models such as BERT Devlin et al. (2019) and GPT Radford et al. (2019), there is a growing interest in extending these models for cross-modal representation learning. Vision-language tasks, such as image captioning, benefit from a larger training corpus, which usually requires tedious human work for the data labelling. The goal of VLP is to learn a cross-modal representation of image-text pairs in a self-supervised manner, which can be adapted to down-stream tasks via fine-tuning Li et al. (2020b). VLP methods commonly use multi-layer self-attention Transformers to learn cross-modal contextualised representations, based on a separate embedding of each modality. A common model architecture and training paradigm in Tan & Bansal (2019); Li et al. (2019a;b); Yu et al. (2019); Lu et al. (2019); Tran et al. (2020); Su et al. (2020); Qi et al. (2020b); Zhou et al. (2020); Chen et al. (2020) is to take visual region features of an image, and the word embeddings of its paired text as input. It relies on the self-attention mechanism to learn image-text alignments and produces cross-modal contextual representations. Luowei Zhou *et al.*Zhou et al. (2020) proposed a unified encoder-decoder model, which can be fine-tuned for both vision-language generation and understanding tasks. However, the VLP models rely heavily on an external pre-trained object detector, which leads to a heavy computational load and require bounding box annotations.

**Attributes, relationships and context.** The task of producing richer and diverse captions by including object attributes, their relationships and the global scene context, defines an important research direction in image captioning. One way to achieve this is through the use of dense captions Krishna et al. (2017) during model training Desai & Johnson (2020). Attribute-driven approaches aim at boosting the performance of a captioning model using the high-level semantic information present in images. Ting Yao *et al.*Yao et al. (2017) are one of the firsts to devise an attribute augmented architecture by integrating the object attributes into a CNN LeCun et al. (1989) plus RNN(LSTM) image captioning framework. Hui Chen *et al.*Chen et al. (2018) also propose an attribute-based approach by developing an attribute-driven attention model Yao et al. (2018).

For vision and language tasks, the authors of Knowledge Augmented Transformer (KAT) Gui et al. (2021) propose a method to solve the Visual Question Answering (VQA) task. KAT focuses on Visual Question Answering (VQA) with a rather complex method that could not be easily adapted for other vision and language tasks. For other NLP tasks involving only text, such as generative commonsense reasoning, there are recent works that explore the idea of integrating external knowledge via knowledge graphs. For instance, KG-BART Liu et al. (2020) used knowledge graphs to provide relational information among the commonsense concepts, which helped them to produce more logical and natural sentences. ERNIE Yu et al. (2021) is a single modality (text) method, which encodes the graph structure of the knowledge graph using only knowledge embedding algorithms like TransE Bordes et al. (2013), while we propose using a knowledge encoder based on Graph Attention Networks (GAT) Veličković et al. (2017) which can capture the richer connectivity structure in the graph.

## 3 METHODOLOGY

The goal of image captioning is to provide a description of an image $I$ in the form of a textual sequence $S$, where $S = \{w_1, w_2, ..., w_n\}$ consists of $n$ words. Existing image captioning models usually consist of an **OD** module to extract regional features ($v$) from the raw image (**I**), and a multimodal module (**MM**) to generate a textual description ($t$). Recent works Li et al. (2020b) show that the object tags ($\alpha$) extracted from the detector can serve as anchoring points across modalities. This procedure can be expressed as follows: $(v, \alpha) = \mathbf{OD}(I), \quad t = \mathbf{MM}(v, \alpha)$.

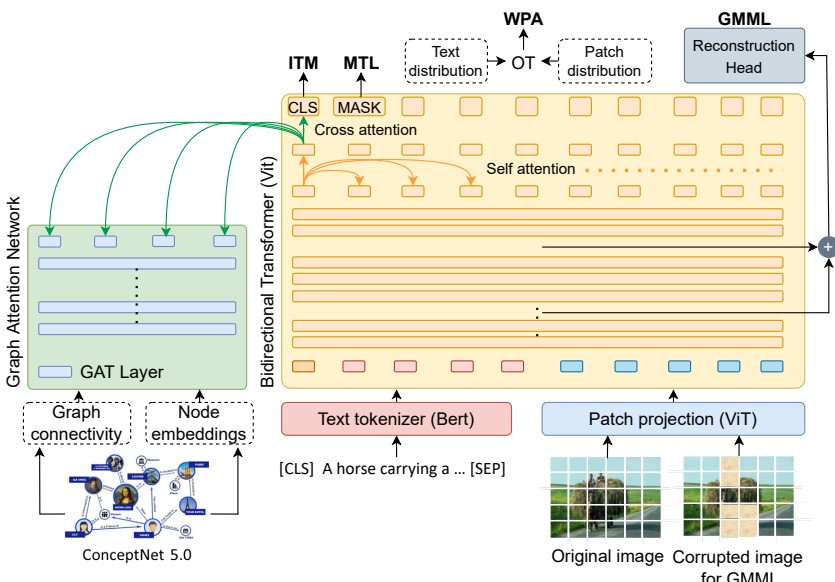

Figure 2: The model architecture for our proposed method. A textual description of the image and the linear projections of image patches are embedded using a ViT Devlin et al. (2019) based Vision and Language encoder. Then relevant external information from the knowledge graph is incorporated to this encoder through cross attention. ITM: Image Text Matching, MTL: Masked Token Loss, GMML: Group Mask Model Learning and WPA: Word Patch Alignment.

These methods usually use pre-trained models to encode the ground truth captions associated with the images based on textual information alone. In doing so, important relations between the semantic concepts, which are not explicitly annotated in the captions, are ignored. In order to overcome this drawback, we propose to incorporate external knowledge into the multimodal model. Specifically, we construct a knowledge graph **G** to embed external information from a knowledge base. An overview of our model architecture is presented in Figure 2. A textual description of the image and the linear projections of image patches are embedded using a ViT Devlin et al. (2019) based Vision and Language encoder. It cannot be overemphasised that we do not need any object detectors to provide pre-computed features into the transformer. We work directly with image patches. The relevant external information from the knowledge graph is incorporated to this encoder through cross attention. In Section 3.1, we explain the construction process of our KG and its formalisation. The method leverages a knowledge graph built from ConceptNet Speer et al.. ConceptNet is a labelled graph which connects words and phrases of natural language connected by edges that denote commonsense relationships between them. Most of the recent Vision and Language (VL) models are based on the Transformer architecture Vaswani et al. (2017). One of the most important components in Transformer is the self-attention mechanism. This involves computing a mapping between a query and a set of key-value pairs to an output, where the query $Q$, keys $K$, values $V$, and output $A$ are all vectors.

$$Q = H_l W_q, \quad K = H_l W_k, \quad V = H_l W_v, \quad A = softmax(\frac{QK^T}{\sqrt{d}})V \quad (1)$$

$H_l \in R^{N \times d}$ is the input hidden vector to the $l-th$ Transformer layer, $W_q, W_k, W_v \in R^{d \times d}$ are projection matrices, $N$ is the sequence length and $d$ is the hidden vector's dimension.

**Image encoding.** Naive application of self-attention to images would require that each pixel attends to every other pixel. With quadratic cost in the number of pixels, this does not scale to realistic input sizes. ViT Dosovitskiy et al. (2020) proposed to reshape the image into a sequence of flattened 2D patches. Following their approach, the input image $I \in R^{C \times Y \times W}$ is sliced into patches and flattened to $v \in R^{J \times (P^2 C)}$ where (Y, W ) is the resolution of the original image, C is the number of channels,

$(P, P)$ is the patch resolution and $J = YW/P^2$ is the resulting number of patches. Followed by linear projection $V \in R^{(P^2C) \times Y}$ and position embedding $V^{pos} \in R^{(J+1) \times Y}$, $v$ is embedded into $v \in R^{J \times Y}$. Patch projection, in contrast with multi-stage object detectors, translates the visual embedding step to the level of textual embedding by tokenization. The text and image embeddings are summed with their corresponding modal-type embedding vectors $t^{type}$, $v^{type}$, and concatenated into a combined sequence $z^0$. The contextualised vector $z$ is iteratively updated through $D - depth$ transformer layers up to the final contextualised sequence $z^D$. The role of GMML is to give consistent representation of data-tokens on an object which helps to overcome the challenges of detector free designs. The absence of commonsense and self-supervised pretraining in SimVLM (ViLT, CPTR) can only be overcome by the use of a larger corpus to outperform detector-based frameworks. For all experiments, we use weights from ViT-B/16 pre-trained on ImageNet. Hidden size is 768, layer depth $D$ is 12, patch size is 16, MLP size is 3072, and the number of attention heads is 12.

### 3.1 KNOWLEDGE GRAPH CONSTRUCTION

In this section, we explain how to construct and learn the embedding of the information conveyed by the knowledge graph from the large commonsense knowledge base ConceptNet 5.0 Speer et al.. We use $\Gamma$ to denote the semantic entities vocabulary, which is the set of all entities extracted from the image captions (objects and actions). The knowledge graph is denoted as $G = (\Upsilon, E, R)$, where $\Upsilon$ is the set of semantic entities, $E$ is the set of edges and $R$ is the set of relations among entities. For a pair of entities $\upsilon_i \in \Upsilon(subject)$ and $\upsilon_j \in \Upsilon(object)$, linked by the relation, $r_{ij} \in R$, the edge $e_{ij} \in E$ can be represented as a triplet $(\upsilon_i, r_{ij}, \upsilon_j)$. Specifically, we assume that the semantic vocabulary is a subset of knowledge graph entities, namely $\Gamma \subset \Upsilon$. Given an unordered set of $k$ commonsense entities $x = \{\gamma_1, \gamma_2, ..., \gamma_k\}$, where each entity $\gamma_i \in \Gamma$ is an object (noun) or action (verb), we want to allow the flow of information through cross attention from the knowledge graph encoder into the multimodal model. Our aim is to boost the performance of the IC task with the help of the commonsense knowledge database $G$, which can be treated as auxiliary information.

More formally, the problem can be stated as follows: $h : \{\Gamma, G\} \rightarrow \{G\}$ that takes the semantic entities sets $\gamma \in \Gamma$ and the knowledge graph $G$ as the input to first learn a commonsense knowledge graph $G_\gamma$, and then $g : \{\Gamma, G_\gamma\} \rightarrow Y$ to generate the final token embeddings with the commonsense knowledge incorporated. Specifically, $G_\gamma \subset G$ consists of all concept triplets $(\upsilon_i, r_{ij}, \upsilon_j)$, where $\upsilon_i$ and $\upsilon_j \in \Gamma$ and $r_{ij} \in R$ is the relation between the semantic entity pair. Most of the semantic entities in image captions correspond to a knowledge graph unigram entity, so we can directly match the semantic entities set to the entities from the knowledge graph to generate $G_\gamma$. We rank the corresponding $G$ nodes for each semantic entity according to the word similarity scores and select their potential top-k similar nodes adding them to $G_\gamma$. To calculate the word similarity scores, we use the pre-trained GloVe Pennington et al. (2014) embedding as the representation of each entity node in the knowledge graph. Since some of the semantic entities do not have a direct connection in the knowledge graph, instead of using $G_\gamma$ directly, we use a knowledge embedding method named TransE Bordes et al. (2013) to learn their entity and relation embeddings. Part of the proposed methodology for extracting the sub-graph of interest is inspired by the one followed in KG-BART Liu et al. (2020). But KG-BART is a single modality (text) processing method. There are also architectural differences. KG-BART needs an encoder-decoder architecture with knowledge injection in two stages, while ours proposes an encoder only model with an external (one-stage) knowledge encoder. The incorporation of external information through cross-attention is more flexible than what KG-BART proposed. Other methods such as Vilt Kim et al. (2021), CPTR Liu et al. (2021) and SimVLM Wang et al. (2021) already introduced a detector free image interpretation method, but (to the best of our knowledge) we are the first one to approach this task by masked image model learning and report the benefits of using the pre-text task introduced by GMML Atito et al. (2021b) in multimodal analysis.

### 3.2 KNOWLEDGE INJECTION

We incorporate external knowledge into the main vision and language model during training by allowing cross attention into a transformer encoder, specifically a GAT model. In the last ViT layer, after the original self-attention module we add a cross-attention module where the queries come from the previous ViT layer while the keys and values come from the output of a corresponding GAT layer, so that the KG encoder's knowledge information can flow into the ViT. Figure 2 shows this

process. The KG encoder architecture follows the GAT model proposed by Veličković et al. (2017). The input to each layer of the GAT is a set of node features, $h = \{h_1, h_2, ..., h_N\}, h_i \in R^F$, where $N$ is the number of nodes, and $F$ is the number of features in each node. The layer produces a new set of node features, $h' = \{h'_1, h'_2, ..., h'_N\}, h'_i \in R^F$, as its output. A shared linear transformation, parametrized by a weight matrix, $W \in R^{F \times F}$, is applied to every node. We then perform self-attention on the nodes. The shared attentional mechanism computes attention coefficients: $e_{ij} = a(Wh_i, Wh_j)$ that indicate the importance of node $j$'s features to node $i$. We inject the graph structure into the mechanism by performing masked attention, that is, we only compute $e_{ij}$ for nodes $j \in N_i$, where $N_i$ is some neighbourhood of node i in the graph. In all our experiments, we use the elements of the first-order neighbourhood of $i$ (including $i$). The connectivity information is extracted from the ConceptNet sub-graph corresponding to each set of nodes, built as explained in Section 3.1.

### 3.2.1 TRAINING OBJECTIVES

Optimizing the alignment between text tokens in the caption $c$ and all image features $v$ is a crucial objective for tasks involving cross-modal representation learning. Contrastive methods to learn an instance-level alignment between the whole image and the caption have been successfully used for this purpose in the past Chen et al. (2020); Li et al. (2020b); Kim et al. (2021). In this work, we specifically use the **Image Text Matching (ITM)** Chen et al. (2020) loss. We randomly sample a set of "polluted" image representations by replacing the aligned image, with probability 50%, with a different image randomly sampled from the dataset. The encoder output on the special token $[CLS]$ is assumed as the aggregated vision-language representation of $(c, v)$. A single linear layer projects the pooled output feature to logits, acting as a binary classifier $f(.)$. Finally, we compute the negative log-likelihood. The output is a binary label $y \in \{0, 1\}$, indicating if the sampled pair is a match. This ITM loss is defined as: $\mathcal{L}_{\mathcal{ITM}} = -\mathbb{E}_{(c,v) \sim \mathcal{D}} log(p(y|f(c, v)))$.

Furthermore, in order to provide a more fine-grained alignment between word tokens and image regions, we use the **Word Patch Alignment (WPA)** Kim et al. (2021) loss. This objective is based on Optimal Transport, which effectively calculates the minimum cost of transporting the contextualized image embeddings from the local patches to the word embeddings (and vice versa). We compute the alignment score between two subsets of $z^D : z^D|t$ (textual subset) and $z^D|v$ (visual subset), using the inexact proximal point method for optimal transports (IPOT) Xie et al. (2020). We set the hyperparameters of IPOT following Chen et al. (2020) and add the approximate Wasserstein distance multiplied by 0.1 to the ITM loss. Learning robust representations for each individual modality (language and text) is another important factor in multimodal tasks. Predicting masked words based on the observation of their surrounding words is an effective task for language modelling Devlin et al. (2018), as it helps to model contextual relationships. We apply the **Masked Token (MTL)** loss as formulated by Li et al. (2020b). Each input token in the caption $c$ is randomly masked with probability 15%, with a maximum of 3 tokens per caption to be masked out. The masked ones are replaced with a special token [MASK]. The model is trained to predict these masked tokens from its contextualized vector $z^D$ and all image features $v$ by minimising the negative log-likelihood: $\mathcal{L}_{\mathcal{MTL}} = -\mathbb{E}_{(v,c) \sim \mathcal{D}} log(p(c_i|c_{\setminus i, v}))$.

For learning useful and robust image representations, self-supervised approaches have reported state of the art results on vision only tasks, e.g. image classification, while being more scalable and interpretable Atito et al. (2021a). We adopt the **Group Mask Model Learning (GMML)** objective as introduced in Atito et al. (2021b). We apply a transformation to local patches of the image in blocks of neighbouring tokens arranged spatially. SiT Atito et al. (2021b) authors showed that it is critical to transform neighbouring tokens in order to extract the context provided by neighbouring patches in the image. In this work, the local transformation applied involved random drop of connected patches. The objective of the image reconstruction is to restore the original image $\mathbf{x} \in \mathbb{R}^{H \times W \times C}$ from the GMML manipulated image $\hat{\mathbf{x}}$. For this task, we use the $\ell 1$-loss between the original and the reconstructed image: $\mathcal{L}(\mathbf{W}) = \sum_k^N \left( \sum_i^H \sum_j^W \mathbf{M}_{i,j}^k \times |\mathbf{x}_{i,j}^k - \bar{\mathbf{x}}_{i,j}^k| \right)$, where

$\mathbf{M}_{i,j} = \begin{cases} 1, & \text{if } \mathbf{x}_{i,j} \text{ is manipulated} \\ 0, & \text{otherwise} \end{cases}$ $\mathbf{W}$ denotes the parameters to be learned during training, $N$ is the batch size, $\mathbf{M}$ is a binary mask with 1 indicating the manipulated pixels, and $\bar{\mathbf{x}}$ is the reconstructed image. The reconstructed image $\bar{\mathbf{x}}$ is obtained by averaging the output features from the intermediate blocks of the transformer encoder $E(.)$ and feeding the output to a light decoder $D(.)$:

$\bar{\mathbf{x}} = D\left(\sum_{b \in \mathcal{B}} E_b(\hat{\mathbf{x}})\right)$. Where $E_b(.)$ is the output features from block $b$ and $\mathcal{B}$ is a pre-defined index set of transformer blocks that are included in the decoding process. In this work, we set $\mathcal{B}$ to $\{8, 10\}$. Furthermore, MTL and GMML losses serve in our architecture two purposes: the aforementioned benefit for individual modality robustness and their contribution to cross modal representation learning. The latter is possible as we allow attention over both modalities while optimizing these losses. The full training objective is:

$$\mathcal{L} = \mathcal{L}_{\mathcal{MTL}} + \mathcal{L}_{\mathcal{ITM}} + 0.1\mathcal{L}_{\mathcal{WPA}} + \mathcal{L}_{\text{GMML}} \qquad (2)$$

## 4 EXPERIMENTS

### 4.1 IMPLEMENTATION DETAILS

We use AdamW Loshchilov & Hutter (2017) optimizer with base learning rate of $10^{-4}$ and weight decay of $10^{-2}$. The learning rate was warmed up for 10% of the total training steps and was decayed linearly to zero for the rest of the training. We resize the shorter edge of input images to 224 and limit the longer edge to under 320 while preserving the aspect ratio. Patch projection of our model, using ViT-B/16 yields $14 \times 20 = 280$ patches for an image with a resolution of $224 \times 320$. We interpolate the learned positional embeddings of ViT-B/16 to fit the size of each image and pad the patches for batch training. We use the *bert-base-uncased* tokenizer to tokenize text inputs. We use a Part of Speech Tagger (POS) Akbik et al. (2019) to detect nouns and verbs in the captions and then these were matched with the top-k nodes from ConceptNet. We pre-train all our model variants for 20 epochs on 4 NVIDIA V100 GPUs with a batch size of 128. For fine tuning on the image captioning task, we train for another 10 epochs.

**Model configurations**. On Table 1 we report the performance of several model configurations. **A (ITM + MTL + GMML)**: here we do not use the WPA loss based on OT but the GMML objective is included for the first time. **B (ITM + MTL + GMML + WPA)**: this configuration uses all the optimization objectives represented in Figure 2. **C (ITM + MTL)**: this variant uses the same procedure introduced by Liu et al. (2020) to inject the external knowledge into the multimodal encoder. Instead of allowing cross attention, the embeddings from the GAT are directly added into the intermediate language tokens on the the ViT based encoder. **D (ITM + MTL)**: this configuration uses the cross attention mechanism to allow the flow of information from the knowledge graph encoder to the multimodal model. **E (ITM + MTL + GMML + WPA)**: in this variant we change the layers used by GMML from the last layer of the multimodal encoder to layers $8^{th}$ and $10^{th}$ combined. **E+20**: we also report the same configuration E but trained for 20 more epochs. We re-implemented the methods in CPTR Liu et al. (2021) and SimVLM Zhang et al. (2021a) following their instructions. The rest of the implementations were the ones released by their authors.

#### 4.1.1 DATASETS

We use the Microsoft COCO Captions Chen et al. (2015) dataset for pre-training and training. No additional data is used. For the COCO Captions dataset, we adopt the widely used Karpathy split Karpathy & Fei-Fei (2015) to conduct our experiments. Specifically, the dataset consists of 113 287 images for training, 5000 images for validation, and 5000 images for testing. Each image has 5 annotated captions with an average caption length of 11.81 ± 2.81 after being tokenized with the *bert-base-uncased* tokenizer. Besides, we report evaluation results on the NoCaps Agrawal et al. (2019) dataset. The NoCaps dataset provides a benchmark with images from the Open Images dataset Kuznetsova et al. (2020) to test models' capability of describing novel objects which are not seen in the training corpus.

### 4.2 RESULTS AND DISCUSSION

We perform comparisons of the proposed method with state of the art models. Following other work in image captioning, we use several widely-used automatic metrics to assess the performance; such as: BLEU Papineni et al. (2002), METEOR Denkowski & Lavie (2014), Rouge Lin (2004) and CIDEr Vedantam et al. (2015). Table 1 presents the experimental results achieved in the proposed model evaluation on the COCO Captions and NoCaps datasets. We can see that our method out-

Table 1: The image captioning evaluation results (single model) obtained on the MSCOCO Captions dataset. All models are trained on the training set of the COCO Captions dataset following the Karpathy split Karpathy & Fei-Fei (2015). B: BLEU@4, M: Meteor, R: Rouge and C: Cider.

| Methods | COCO Captions | | | | NoCaps | | | |
|---|---|---|---|---|---|---|---|---|
| | B@4 | M | R | C | B@4 | M | R | C |
| Up-Down Anderson et al. (2018) | 28.5 | 20.4 | 48.3 | 111.5 | 19.2 | 23.0 | 50.9 | 54.3 |
| Oscar Li et al. (2020b) | 34.3 | 27.6 | 54.7 | 115.2 | 21.0 | 24.8 | 53.1 | 57.2 |
| SimVLM Wang et al. (2021) | 32.9 | 26.5 | 53.9 | 110.2 | 20.5 | 24.6 | 52.0 | 55.1 |
| CPTR Liu et al. (2021) | 31.3 | 25.2 | 53.1 | 108.6 | 19.8 | 24.0 | 51.2 | 54.9 |
| VilT Kim et al. (2021) | 33.0 | 26.8 | 54.1 | 113.3 | 20.9 | 24.0 | 52.7 | 56.0 |
| Oscar (with KG) | 35.8 | 28.3 | 56.0 | 116.1 | - | - | - | - |
| Ours (A: ITM + MTL + GMML) | 30.2 | 25.0 | 51.0 | 112.7 | - | - | - | - |
| Ours (B: ITM + MTL + GMML + WPA) | 33.7 | 27.0 | 54.5 | 114.2 | - | - | - | - |
| Ours (C+KG [KgBart-like]:ITM + MTL) | 34.8 | 27.9 | 55.2 | 116.0 | - | - | - | - |
| Ours (D+KG: ITM + MTL) | 35.7 | 28.3 | 56.0 | 118.5 | - | - | - | - |
| Ours (B+KG: ITM + MTL + GMML + WPA) | 36.1 | 29.0 | 56.2 | 118.5 | - | - | - | - |
| Ours (E: ITM + MTL + GMML + WPA) | 36.5 | 29.4 | 56.8 | 118.5 | - | - | - | - |
| Ours (E+20: ITM + MTL + GMML + WPA) | **38.1** | **30.3** | **57.9** | **121.5** | **22.3** | **26.7** | **55.2** | **59.1** |
| $\Delta$ difference b/w best SOTA and our E+ | ↑ **3.8** | ↑ **2.7** | ↑ **3.2** | ↑ **6.3** | ↑ **1.3** | ↑ **1.9** | ↑ **2.1** | ↑ **1.9** |

Table 2: The image-text retrieval evaluation results obtained on the test set of MSCOCO Captions dataset. All models are trained exclusively on the MSCOCO training set using the Karpathy split Karpathy & Fei-Fei (2015).

| Methods | Rank@1 | Rank@5 | Rank@10 |
|---|---|---|---|
| Oscar Li et al. (2020b) | 35.1 | 66.2 | 76.8 |
| Ours (E+) | **36.5** | **67.4** | **77.9** |
| $\Delta$ difference b/w best SOTA and our E+ | ↑ **1.4** | ↑ **1.2** | ↑ **1.1** |

performs the baseline by a significant margin in most of the metrics. The improvement is observed more clearly when measured by the CIDEr score with a 3.3% improvement. Table 2 shows the text retrieval results on the 1k test set of MSCOCO. Following the Oscar Li et al. (2020b) methodology, during training, we formulate this task as a binary classification problem. Given an aligned image-text pair, we randomly select a different image or a different caption to form an unrelated pair. The final representation of the [CLS] token is used as the input to a binary classifier to predict whether the given pair is aligned or not. We decided to use the binary classification loss instead of the ranking losses following the findings of Oscar Li et al. (2020b) and Imagebert Qi et al. (2020a), which reported a better performance for the binary loss. In the testing stage, the probability scores are used to rank the given image-text pairs of a query. Following Li et al. (2020a), we reported the top-K retrieval results on the MSCOCO test set. We included results on image-text retrieval to show how CLMIU is able to perform other vision and language tasks. The improvement over the baseline shows that incorporating external knowledge also boosts the performance of image-text retrieval. The results on the NoCaps dataset show that the model generalizes to unseen object categories. The size of the images for this task was the same as in all previous image captioning experiments (224x320). We anticipate that using bigger image size, e.g. 640x480, would have a positive impact on performance.

The results suggest that incorporating external commonsense knowledge boosts the performance of the VLP model. The fact that the improvement is more apparent in the CIDEr metric owes to the nature of this metric. It takes into account the semantic correctness and quality of the generated

captions better. The other metrics were adapted from pre-existing NLP tasks and as such they are more focused on analysing the presence of overlapping n-grams between the candidate and reference sentences. An n-gram overlap is neither necessary nor sufficient for two sentences to convey the same meaning or the same semantic correspondence.

### 4.2.1 LIMITATIONS

There are several potential limitations of CLMIU that will be addressed in future work, including: 1) exploring dynamic knowledge representations that can be adjusted by feedback during training; and 2) reducing the need for paired data during training by leveraging language and vision only pre-training and/or self-supervised learning. The curation process involved in the creation of the original COCO captions dataset could cause the models trained on it to reproduce undesirable social biases. For this reason, mitigation measures should be employed before any production deployment of this method. Furthermore, we advise against the use of CLMIU or any its variants in automated scenarios where it could influence critical decisions with consequences for human beings. We do not advocate the use of CLMIU as part of any mass surveillance system.

### 4.2.2 ABLATIONS

The following ablation study aims to investigate the relative contributions of architectural and training decisions to the final model performance.

**Into which layers is best to inject the KG information?:** adding a cross attention module after every layer on the ViT to allow information flow from the KG, does not lead to significant improvements. We argue that: the concepts and relationships encoded in the KG sit on the top of the abstraction hierarchy. Therefore, this information is better exploited and integrated by the multimodal ViT encoder in its final layers. **Do existing methods improve by just coupling the KG encoder?:** the cross attention module allows to couple the KG encoder into any Transformer-based multimodal encoder. We retrained existing IC SOTA methods, such as: Oscar Li et al. (2020b), after incorporating the KG encoder through cross attention. The improvements reported show that they do benefit from this external information. The fact that the image encoder is a frozen pre-trained object detector limits the overall gain. We expect that other similar SOTA models (e.g. VinVL Zhang et al. (2021a)) will also benefit from this strategy. **GMML objective leads to better image representations:** previous attempts Fang et al. (2021) to use object detector free IC models had to incorporate additional training objectives designed to model the objects in the image. SiT Atito et al. (2021b) and MC-SSL0.0 Atito et al. (2021a) succeeded in modelling multiple concepts present in an image through self-supervised pre-training. They showed that these learned concepts or object categories are consistent across different images and across instances of the same category. We also found that this training strategy boosted performance and we attributed it to better image representations being learned. **Effect of Longer Training:** as reported in supplementary material Appendix Figure 3, increasing the number of training epochs causes the performance to steadily increase. This results suggest that the approach is scalable with respect to training time available.

## 5 CONCLUSIONS

We addressed the problem of image captioning and presented two major contributions. The first is the proposed methodology for incorporating commonsense knowledge into the captioning process to boost the performance of the learnt vision-language model. This has been achieved by injecting the semantic relations among concepts conveyed by a knowledge graph using a transformer architecture. The experimental results confirm that this auxiliary information improves the performance of existing image captioning systems as well as the novel architecture proposed in the paper. The key innovation of the proposed image captioning system is its object detector free design, which has been enabled by a new method of vision transformer training using the Group Masked Model Learning (GMML). This revolutionary pre-training method has major advantages, namely: **a)** enabling end to end training; **b)** requiring no bounding box annotation of training images, and most importantly; **c)** greater efficiency in extracting contextual information, resulting in much better image representation. The image captioning system based on the proposed architecture and incorporating the common sense knowledge has defined a new state-of-the-art performance on the COCO Captions dataset.

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

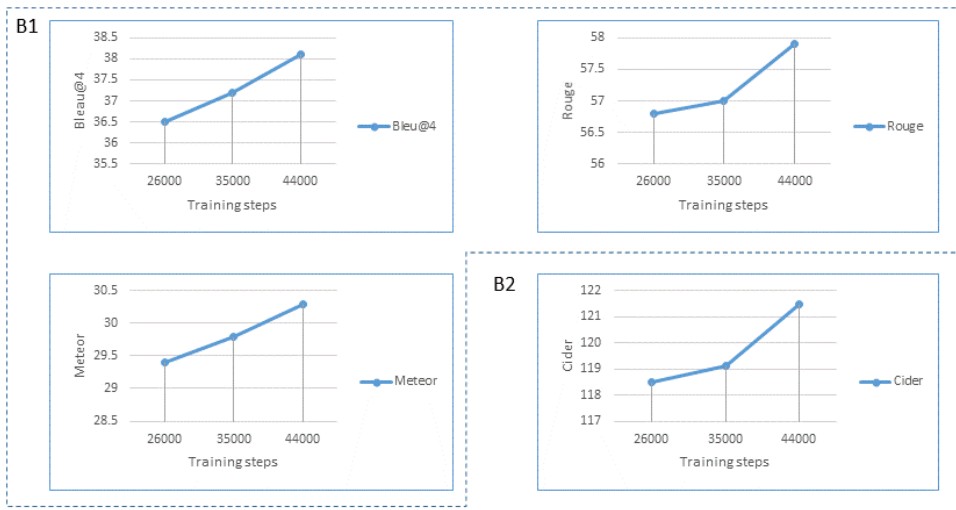

Figure 3: Effect of longer training on performance improvement.

## A   APPENDIX

**Appendix A.1** shows an analysis of CLMIU's attention patterns, where it can be seen that image regions attend to text tokens that are conceptually related. **Appendix A.2** shows the effect of longer training on the performance of the model. **Appendix A.3** shows visualisations of the self-attention heads in the last layer of the model for different images.

### A.1   CROSS-MODAL ALIGNMENT

In Figure 4, the transportation plan learned by the Word Patch Alignment objective is manifest in a heatmap for each of the text tokens highlighted in red colour. Each square tile represents a patch, and its opacity indicates how much mass is transported from the highlighted word token. The figure shows how specific objects are aligned with their textual identifiers. This is the case of the *candles* in image *a* or the *ball* in image *c*. Abstract concepts are shown to be distributed over the objects involved in their definition. In image *a* the concept of *birthday* is connected with most of the objects related to it. Besides, in image *b* the concept of *riding (a bicycle)* attends to a similar area than the term *bicycle*, as these are semantically related.

### A.2   EFFECT OF LONGER TRAINING

Figure 3 shows that increasing the number of training steps produces a steady increase on performance in all of the metrics reported. Besides, this also shows that training metrics still do not saturate and they should continue to improve with longer training time. The longer training benefits classic metrics (Figure 3 B1) and more semantic metrics (Figure 3 B2) as well.

### A.3   SELF-ATTENTION HEADS

Figure 5 shows the self-attention heads from the last layer. We visualise the attention maps computed using the [CLS] token as a query for the different heads in the last layer. We found that the main objects or parts of the scene, which are necessary for generating a description, are identified and preserved across the attention heads.

a) A woman blowing out candles on a birthday cake.

b) A woman and a child riding a bicycle with a dog behind them.

c) A baseball player swinging at a ball during a game.

d) A cat is sleeping on a laptop.

Figure 4: Visualizations of the transportation plan for the Word Patch Alignment. The images are from a) the internet, (b, c and d) MSCOCO dataset.

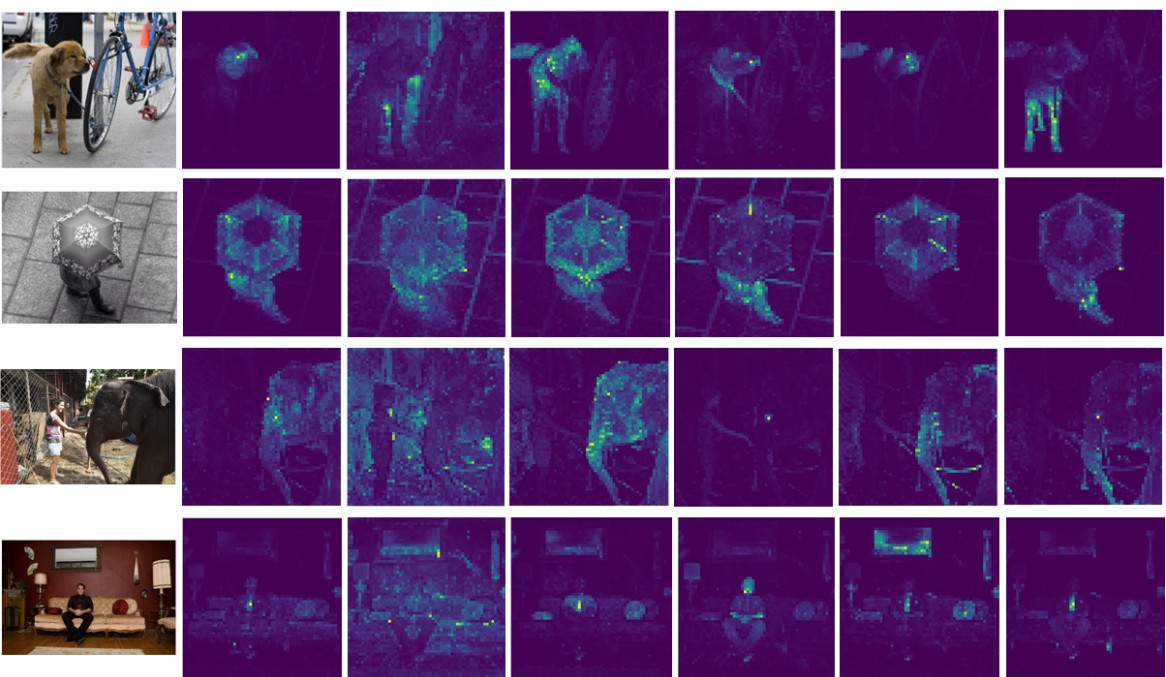

Figure 5: Self-attention heads from the last layer.

