# OpenReview forum: "CLMIU: Commonsense Learning in Multimodal Image Understanding."
_ICLR.cc/2023/Conference — Submitted to ICLR 2023_

### Official Review · Reviewer_C2N5 · 2022-10-23

**Confidence:** 3
**Correctness:** 2
**Technical Novelty And Significance:** 2
**Empirical Novelty And Significance:** Not applicable
**Recommendation:** 5

**Clarity, Quality, Novelty And Reproducibility:**

The paper is well written and motivated. The  experimental results are not clear. Why is experiment E - not E+KG, assuming KG means adding knowledge graph to the pretraining.

**Strength And Weaknesses:**

Strengths -
The motivation and intuition behind incorporating external commonsense knowledge during pretraining is interesting .The authors also use  a detector free approach to image modeling thus training an end to end system.

Weaknesses -

The role of GMML is not very well experimentally evaluated. What happens if GMML is not used at all? And the only losses used are ITM + MTL + WPA ? Is it not very convincing what advantages it brings when knowledge is incorporated .

Qualitative visualization of what commonsense knowledge nodes were important for the downstream task ?

The contributions of the paper are not significant and not well illustrated in experimental results.



**Summary Of The Paper:**

The authors propose to pretrain a vision language transformer with external commonsense knowledge to improve on downstream tasks where external knowledge is relevant. They validate their approach on the downstream task of image captioning and image retrieval and show improvement when the model is trained with commonsense knowledge.

**Summary Of The Review:**

The paper proposes an interesting idea but the contributions are not sufficient and the experimental results do not fully convey the addition of commonsense knowledge and its effetcs. Other downstream tasks such as VCR or VQA might be more reasonable the effect of commonsense knowledge on pretraining.

---

> ### Author Response · Authors · 2022-11-16
> **Response to Reviewer C2N5**
>
> **The role of GMML is not very well experimentally evaluated.**
>
>  A comparison between the results of Ours (B: ITM + MTL + GMML + WPA) in Table 1 with the ones reported for Vilt, shows the benefits of adding GMML. Vilt has a similar base architecture to our model when you do not include our Knowledge Encoder. The optimization objectives used on Vilt are ITM + MTL + WPA.
>
> **Qualitative visualization of what commonsense knowledge nodes were important for the downstream task?**
>
> We agree these visualizations are necessary to fully understand how the knowledge propagates from the KG into the multimodal model. Knowledge localization and editing is a complex task with ongoing research. We plan to explore the best alternatives to include in our model.

---

### Official Review · Reviewer_3AwF · 2022-10-23

**Confidence:** 4
**Clarity, Quality, Novelty And Reproducibility:** 1.	Clarity
**Correctness:** 3
**Technical Novelty And Significance:** 2
**Empirical Novelty And Significance:** 2
**Recommendation:** 3

**Strength And Weaknesses:**

Strengths:

1.	This paper proposes to leverage useful information from an external knowledge graph for identifying underlying semantic concepts not represented in the images for caption generation.

2.	The proposed method has achieved better results in image captioning by training on a relatively small dataset compared with several state-of-the-art models of vision-and-language pre-training.

Weaknesses:

1.	It is unclear that the proposed model should be considered as a captioning model or a vision-and-language pre-training model. If it is a captioning model, it should be compared with state-of-the-art baselines of image captioning. Otherwise, it should be evaluated on more cross-modal understanding tasks like the other pre-training models. However, none of these is included in the paper.

2.	The idea of leveraging external knowledge graph for cross-modal understanding tasks is not new and has been adopted in other works like [R1-2]. The authors should discuss the differences between the proposed method and these works. Moreover, for me, the key idea between KG-BART and the proposed method is almost the same except that KG-BART is for NLP tasks.

[R1] Relational Reasoning using Prior Knowledge for Visual Captioning, arxiv 1906. 01290v1.
[R2] Concept Propagation via Attentional Knowledge Graph Reasoning for Video-Text Retrieval, ACM MM 2022.

3.	For efficiency, it seems unpractical to cross attention over all the nodes in the knowledge graph in the last layer of ViT.


**Summary Of The Paper:**

This paper proposes to identify semantic concepts that are not explicitly represented in the given image for caption generation by incorporating knowledge from an external knowledge graph (e.g., ConceptNet). Specifically, a ViT is trained to extract multimodal representations from image-text pairs with multiple pre-training objectives (i.e., Image Text Matching, Word Patch Alignment, Masked Toke Loss and Group Mask Model Learning), and the external information from a pre-trained knowledge graph built on the basis of ConceptNet is injected into the last layer of the ViT through cross attention. Compared with state-of-the-art models of vison-and-language pre-training, the proposed method achieves better captioning performances on MSCOCO.

**Summary Of The Review:**

I lean to negative for the reasons as follows:

1) The idea of leveraging external knowledge graph for multimodal understanding tasks is not new and has been adopted in previous works.

2) No new architecture designs or training strategies are proposed in the paper.

3) Though the authors claim that the proposed model is a vision and language model which can learn multimodal image representations from image-text pairs, evaluations on more multimodal understanding tasks are not included in the paper.

---

> ### Author Response · Authors · 2022-11-16
> **Response to Reviewer 3AwF**
>
> **It is unclear that the proposed model should be considered as a captioning model or a vision-and-language pre-training model. If it is a captioning model, it should be compared with state-of-the-art baselines of image captioning. Otherwise, it should be evaluated on more cross-modal understanding tasks like the other pre-training models. However, none of these is included in the paper.**
>
> We think separating “captioning models” from “vision-and-language pre-training models” is not applicable for most recent research on this topic. Our approach is a transformer-based model that processes multi-modal (images + text) inputs and learns unified feature representations. These learned features could be used for any downstream task that involves vision and language understanding. Examples of these: image captioning, visual question answering, visual commonsense reasoning and image-text retrieval. We included in the comparison strong SOTA models such as SimVLM. Evaluating the model in other tasks would provide stronger evidence of its capabilities, and we plan to do so in the future.
>
> **Novelty and existing work.**
>
> We already discussed the KG-BART approach in the paper and the main differences between their model and ours. KG-BART  is a single modality (text) method which needs an encoder-decoder architecture with knowledge injection in two stages, while ours proposes an encoder-only model with an external (one-stage) knowledge encoder. Besides, the incorporation of external information through cross-attention is more flexible than what KG-BART proposed.
>
> The approach used in [R1] is based on LSTM and GCN. It has been already demonstrated the superiority of Transformer models vs LSTM model for sequence modelling and the same applies to GAT vs GCN for graph modelling. [R2] introduces a method for video text retrieval, this is a different area of research and the architecture and components are different and would not be applicable to the tasks discussed in our paper. The idea of injecting external knowledge is not novel and we reference some relevant and recent approaches in Section 2.
>
> [R1] Relational Reasoning using Prior Knowledge for Visual Captioning, arxiv 1906. 01290v1.
>
> [R2] Concept Propagation via Attentional Knowledge Graph Reasoning for Video-Text Retrieval, ACM MM 2022.
>
> **For efficiency, it seems unpractical to cross attention over all the nodes in the knowledge graph in the last layer of ViT.**
>
>  In Section 3.1 we explain the pruning process involved in extracting subgraphs of interest from the Conceptnet 5.5 knowledge graph. We are allowing cross attention only to a specific subgraph that contains relevant knowledge for a given training example.

---

### Official Review · Reviewer_hmfQ · 2022-10-28

**Confidence:** 4
**Correctness:** 2
**Technical Novelty And Significance:** 2
**Empirical Novelty And Significance:** 2
**Recommendation:** 3

**Clarity, Quality, Novelty And Reproducibility:**

Novelty: the novelty is limited because most of the components are borrowed from existing papers.
Reproducibility: the authors provide many implementation details; it is likely that it can be reproduced by an expert in the field.

**Strength And Weaknesses:**

Strengths:
- The model design is simple and intuitive.
- The proposed method does not require a pretrained object detector.
- The authors provide some ablation studies.

Weaknesses:
- Experiments:
    - No qualitative results.
    - No analysis of how this added knowledge actually increases the "common sense" of the captioning model.
    - Some of the ablation experiments results are only described but not shown in any table. For example, the VinVL + knowledge graph result.
    - The authors only show performance on two tasks, whereas, for such a general vision language model, it is important to see the generalization ability on many different vision language tasks.
    - The result table of image caption retrieval is too simple. Should include more baselines and ablations.
    - The authors say the experiments show that GMML is helpful however, if we look at Table 1, for all methods with GMML, there is no method without GMML to compare to. This seems suspicious to me.
    - The numbers in the tables are weird and the baselines’ scores are far lower than the numbers reported in the original papers. The SOTA performance is lower than what was 5 years ago.
- Paper presentation
    - I am seeing many superscripts in which I am expecting footnotes. For example, for method C, and Oscar in Table 1. However, I don’t see any footnotes. Is that expected?

**Summary Of The Paper:**

The authors propose a Vision language model that can use the external knowledge graph ConceptNet. The model consists of a transformer model with a graph-based model to embed graph information. The authors claim the proposed model can outperform state-of-the-art models in image captioning.


**Summary Of The Review:**

There are many problems in this paper as I described in the weaknesses, thus I suggest reject.

---

> ### Author Response · Authors · 2022-11-16
> **Response to Reviewer hmfQ**
>
> **More qualitative results and analysis of how this added knowledge actually increases the "common sense" of the captioning model.**
>
> We will include in a revision of the paper more qualitative comparisons of the captions generated by our model vs the ones generated by the baselines. We agree that more in-depth qualitative experiments would be useful to show how the external knowledge encoded in the KG’s nodes and relationships benefits the multimodal model. However, Figure 4, Appendix A1 shows how specific objects are aligned with their textual identifiers; while abstract concepts are shown to be distributed over the objects involved in their definition. This visualization provides some qualitative grounding evidence on the cross-modal representations learned by the model.
>
> **Some of the ablation experiments results are only described but not shown in any table. For example, the VinVL + knowledge graph result.**
>
> We mentioned VinVL (Oscar+) as it is an incremental update of Oscar, with an improved object detector. Thus, as incorporating external knowledge benefits Oscar model, it should improve the performance of VinVL as well. We will fix the writing on that section to clarify this.
>
>
> **Additional vision language tasks.**
>
> We plan to include experimental results on additional vision and language tasks in the future.
>
> **Model variant with GMML vs without GMML.**
>
> A comparison between the results of Ours (B: ITM + MTL + GMML + WPA) in Table 1 with the ones reported for Vilt, shows the benefits of adding GMML. Vilt has a similar base architecture to our model when you do not include our Knowledge Encoder. The optimization objectives used on Vilt are ITM + MTL + WPA.
>
> **Reported results compared to SOTA.**
>
> The reviewer is right, the numbers in the tables are lower than the numbers reported in the original papers. This happens because all the baselines and our models were pretrained, trained and tested on data from the Karpathy split of the COCO Captions dataset with no extra V-L pretraining outside COCO captions. The scores reported in the original papers are the result of training on a larger corpus (e.g. 4M images-text pairs) and/or longer training. We are not claiming SOTA performance, but improvement over strong baselines compared on fair grounds. Besides, all variants of our model (A to E) are initialized from ViT-B/16 pre-trained on ImageNet-21k as released by ViT authors. We also used the pretrained bert-base-uncased tokenizer to tokenize text inputs. This is clearly stated in the paper in Section 4.1.1.
>
> **Presentation issues**
>
> Thanks for these; we will fix them in revision.

---

### Official Review · Reviewer_yTLW · 2022-11-05

**Confidence:** 4
**Correctness:** 3
**Technical Novelty And Significance:** 3
**Empirical Novelty And Significance:** 3
**Recommendation:** 6

**Clarity, Quality, Novelty And Reproducibility:**

I appreciate the ablation studies in table 1, but it is not explained in detail in the experiment sessions. E.g., I am curious to know how sensitive the model is to the quality of the knowledge graph. What about we train a transformer based text model and feed their embeddings to the model? Will it achieve similar results?


**Strength And Weaknesses:**

 - The paper is clear overall.
 - The authors provides good ablation study in table 1 to validate their major claim.
 - It is interesting and novel to me to incorporate external knowledge to image captioning tasks, and achieve competitive results.


**Summary Of The Paper:**

This paper presents a model for image captioning, utilizing external knowledge with a graph-based model. The full model contains a transformer (a text tokenizer + a ViT based patch tokenizer), and a graph attention network to get the external knowledge from conceptnet 5.0. The paper claims to achieve state-of-the-art performance on smaller dataset, and can generalize to unseen object categories.


**Summary Of The Review:**

I think the paper passed the threshold for acceptance, based on its novelty.

---

> ### Author Response · Authors · 2022-11-16
> **Response to Reviewer yTLW**
>
> **How sensitive the model is to the quality of the knowledge graph?**
>
> We consider ConceptNet 5.5 to be enough for studying how incorporating external knowledge would benefit multimodal tasks and specifically the image captioning task. Its knowledge is manually collected from many sources that include expert-created resources and crowd-sourcing, among others. We noticed that collecting the triplets in two-hop paths and three-hop paths between each concept pair does improve the overall performance. This may be related to having a subgraph which can provide richer and wider information about the entities and their relationships. Limited experiments were performed on this, due to the impact on computational resources usage as the subgraphs grow.
>
> **What about we train a transformer based text model and feed their embeddings to the model? Will it achieve similar results?**
>
> There are several ways you could replace the Knowledge Encoder by a transformer based text model. For instance, one could build sentences by concatenating triplets of entities and relations in the graph and train a transformer on this corpus. Using this approach could provide some benefit but, the capacity of language models to capture entity relationships is limited when compared to Graph Attention Networks.

---

### Decision · Program_Chairs · 2023-01-20

**Decision:**

Reject

**Justification For Why Not Higher Score:**

N/A

**Justification For Why Not Lower Score:**

N/A

**Metareview: Summary, Strengths And Weaknesses:**

This paper proposes a new Vision language model that combines the strengths of Transformers with graph-based models to convey external common sense knowledge. The motivation and intuition of the paper are interesting. However, the reviewers had many concerns about the contributions, experiments, ablation studies, and analysis of the results. The authors addressed some of them but the reviewers are not fully convinced.